# Prevalence of CMV, EBV, HPV, and HSV among South Asian healthy population: A systematic review and meta-analysis

**Akash Ahmed**[ID]*, **Zwad Al Saiyan**[ID], **Rifa Tamanna Subarna**[ID], **Nafisa Mehreen Naser**, **Nabila Khan**, **Badhan Bhattacharjee**[ID], **Nadia Sultana Deen**

Microbiology Program, Department of Mathematics and Natural Sciences, BRAC University, Dhaka, Bangladesh

* akash.ahmed@bracu.ac.bd

## Abstract

Cytomegalovirus (CMV), Epstein-Barr virus (EBV), Human papillomavirus (HPV), and Herpes simplex virus (HSV) are DNA viruses which are highly prevalent among the general population. Although the prevalence of each of these viruses has been studied separately within South Asian populations, there are no studies regarding the pooled prevalence of these viruses among healthy individuals across South Asia. A systematic search was performed using three databases (PubMed, Scopus, and Cochrane Library) and one search engine (Google Scholar) for original studies on the South Asian population (published from 2000 to 2025). Following the search, DerSimonian-Laird random effect meta-analysis was performed to calculate the overall prevalence of CMV, EBV, HPV, and HSV in South Asia. Based on our eligibility criteria, we found 94 studies from 7 South Asian countries comprising 162,659 healthy individuals. The overall pooled prevalence of the four viruses was 20% [95% CI: 16% to 24%]. The prevalence of the studies ranged from 0% to 100% indicating a significant amount of heterogeneity ($I^2 = 100\%$; $p < 0.01$). The highest pooled prevalence was of CMV (57%; 95% CI: 21% to 89%) followed by EBV (17%; 95% CI: 5% to 34%), HPV (13%; 95% CI: 10%. to 16%), and HSV (9%; 95% CI: 16% to 12%). Furthermore, country-wise analysis showed India to have the majority of the studies. Our findings revealed that 20% [95% CI: 16% to 24%] of healthy individuals who lived in different South Asian countries are infected with one of these DNA viruses, emphasizing the widespread impact across different geographical regions. As these infections can lead to severe health complications, it is crucial to establish preventive guidelines and spread awareness among the healthy population.

**Data availability statement:** All relevant data are within the paper and its Supporting Information files.

**Funding:** The authors received no specific funding for this work.

**Competing interests:** The authors have declared that no competing interests exist.

## Introduction

Cancer is the leading cause of death globally, where viruses account for 13–20% of all cancer cases [1,2]. While Epstein-Barr virus (EBV) and Human Papillomavirus (HPV) are established oncoviruses [2], Herpes simplex virus (HSV) and Cytomegalovirus (CMV) are also considered risk factors for cancer [3,4]. These viruses can directly spread through bodily fluids such as saliva, blood, semen, and breast milk [5–8]. Primary infection with these viruses might be asymptomatic but they can integrate their genetic material into the host chromosome, leading to potential reactivation at any point [9,10]. This suggests a pressing need for monitoring the prevalence of these viruses among healthy individuals.

Viral cancers typically take 15–40 years to develop, rather than emerging immediately after infection [11]. However, the association between viral infections and cancer development has become increasingly evident through extensive research [12–14]. For instance, studies have found a high prevalence of CMV in breast, colon and prostate cancer, as well as in hepatocellular carcinoma [12]. Similarly, EBV has been associated with epithelial cancers and a number of lymphoid malignancies such as Burkitt's lymphoma, Hodgkin Lymphoma, and nasopharyngeal carcinoma [13,14]. It is estimated that EBV is responsible for over a quarter million cases of cancer every year and nearly 2% of all cancer-related deaths are due to EBV-attributed malignancies [15]. In recent years, the connection between HPV, cervical cancer, and several types of squamous cell carcinomas has been well recognized [16]. Studies have found HPV DNA in over 95% of cervical cancers which are the third leading cause of cancer-related deaths among women globally [17]. HSV could be a risk factor for cancer as well since research suggests a potential role of HSV in cervical cancer, where it initiates the oncogenic process [18]. Moreover, according to a Mendelian randomization study, a significant association has been found between HSV infection and an increased risk of head and neck cancer (HNC) [19].

Apart from cancer, these viruses have been linked to neurodevelopmental issues. Viruses, such as CMV [20] and HSV [21], can cross the placental barrier during pregnancy, potentially disrupting brain development and increasing the risk of conditions like autism spectrum disorder (ASD). Moreover, HPV has been found in placental trophoblasts [22,23], and its presence has been associated with a greater likelihood of ASD [24]. Similarly, EBV has been connected to neurodevelopmental impairments. Furthermore, a number of study reports showed that pediatric patients with multiple sclerosis, often positive for EBV, are more likely to experience cognitive challenges compared to their healthy peers [25–27].

The Centers for Disease Control and Prevention (CDC) and The World Health Organization (WHO) have reported a high prevalence of CMV, EBV, HPV, and HSV among the general population [28–31]. However, these prevalence rates differ among geographic regions. For example, in the United States, the prevalence rates are 50% for CMV [32], 66.5% for EBV [33], 40% for HPV [34], and 12.1-48.1% for HSV [35,36] whereas Africa shows notable variation, with lower rates for EBV (20%) [37] and HPV (2–45%) [38] but higher rates for HSV (37.3-90%) [39,40], and CMV (81.8%) [41].

Although extensive prevalence data for CMV, EBV, HPV, and HSV are available for the aforementioned regions, comprehensive prevalence studies for these four viruses are limited in South Asia.

South Asia consists of eight countries: Afghanistan, Bangladesh, Bhutan, India, Maldives, Nepal, Pakistan, and Sri Lanka [42]. These nations collectively have a population of approximately 2.4 billion people, accounting for more than one-fourth of the global population [43]. Various studies have highlighted the prevalence of CMV, EBV, HPV, and HSV in these populations [44–46]. For example, HPV prevalence among cervical cancer patients in India is 94% [44], while EBV prevalence among Hodgkin Lymphoma patients in Bangladesh is 68.1% [45]. Furthermore, among pregnant women in India, CMV and HSV prevalence rates are 8% and 6%, respectively [46]. The positive prevalence of these viral infections among these populations could indicate the presence of latent oncogenes, which may expose many individuals to an increased risk of cancer and ASD. Therefore, a synthesis of these prevalence data would be important for treating asymptomatic CMV, EBV, HPV, and HSV infections as persistent contributors to the emergence of cancer-causing mutant genes rather than as latent infections.

This systematic review and meta-analysis aimed to estimate a pooled prevalence of CMV, EBV, HSV, and HPV among the healthy population of South Asia. Meta-analysis was done to explore the different prevalence rates of the four viruses. Country wise prevalence were also examined for a comprehensive understanding of the distribution and potential demographic disparities in the region. This prevalence data is crucial for informing policymakers and healthcare professionals to implement necessary measures aimed at reducing the transmission and risk associated with these DNA viruses.

## Methods

We followed the latest recommendations and guidelines from the Preferred Reporting Items for Systematic Review and Meta-Analysis (PRISMA 2020) as shown in Fig 1 [47]. Additionally, this systematic review was registered in PROSPERO (CRD42024510467).

### Data source and search strategy

A preliminary search was conducted by one of the co-authors (AKS) to conceptualize the idea and identify the knowledge gap. Afterward, the eligibility criteria were developed by all the co-authors. Three co-authors (RTS, ZAS, and NMN) independently searched for articles indexed in PubMed, Scopus, Cochrane Library, and Google Scholar by following predefined eligibility criteria. The initial search was conducted between January 7, 2024 and March 12, 2024, while the most recent search was performed from May 7, to May 15, 2025. In addition, grey literature was also searched systematically through OpenGrey, WHO, CDC, and the National Institute for Health Research (NIHR) to capture potentially relevant studies. All retrieved records were screened at the title/abstract level by at least two reviewers, followed by full-text screening, with disagreements resolved through discussion with other co-authors. Searches on Google Scholar were performed after logging out from all Google accounts to minimize personalization bias. The search strategy aimed to retrieve articles published between 2000 and 2025, focusing on the prevalence of CMV, EBV, HPV, and HSV among healthy individuals in South Asian countries.

Our search terms included combinations of keywords [e.g., "Herpes simplex virus," "Human Papillomavirus," "Cytomegalovirus," "EBV," "HSV", "HPV", "CMV", "Seroprevalence", "Incidence," "Prevalence," "Frequency" "Distribution", "Healthy", "South Asia," "Bangladesh," "India," "Pakistan," "Nepal," "Sri Lanka," "Bhutan," "Maldives," and "Afghanistan"] to cover all the relevant articles as per our eligibility criteria. The detailed search strategy, list of original keywords, and alternative terms used in this study are given in the S1 Table.

### Eligibility criteria

While developing the search strategy, we observed that abstracts from several articles reported only basic prevalence estimates, whereas essential methodological details such as participant selection criteria and diagnostic methods were

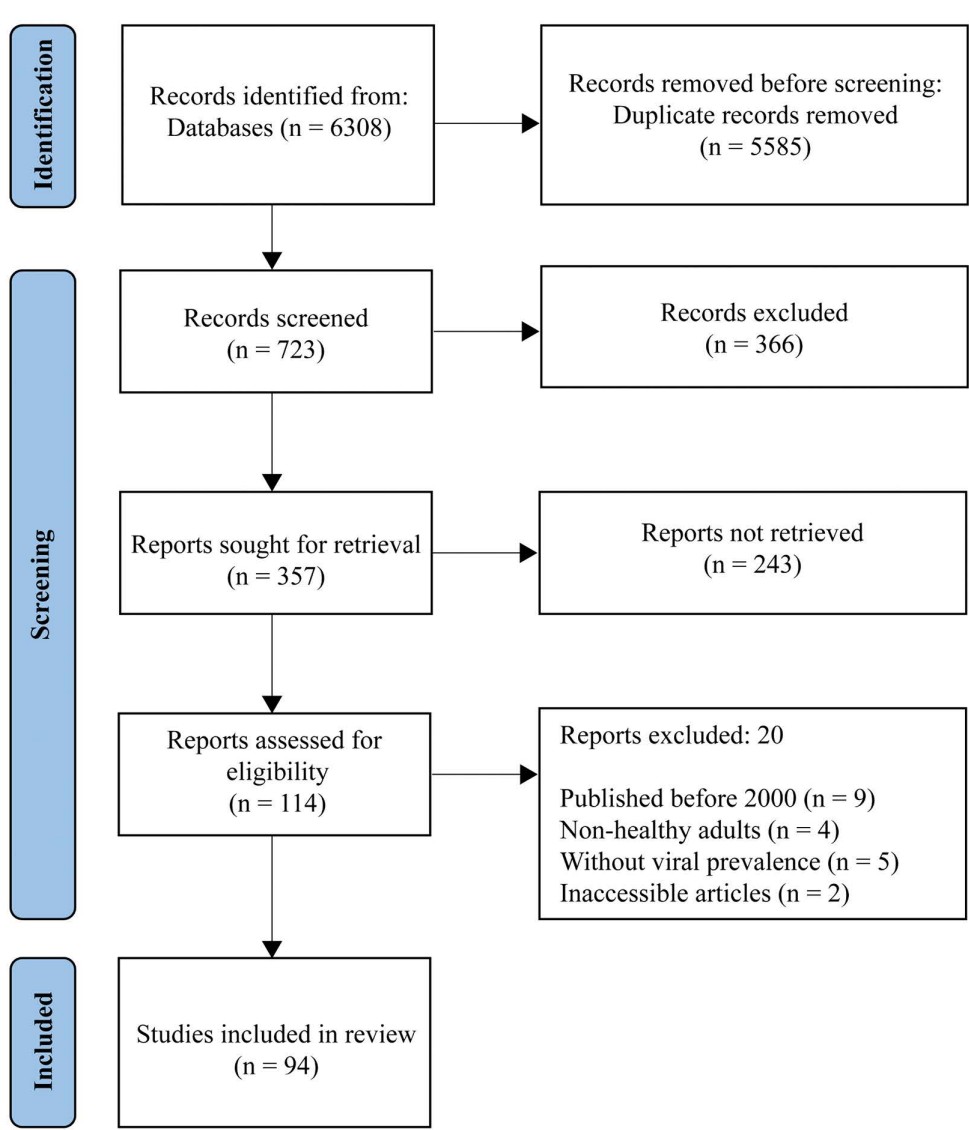

**Fig 1. PRISMA flowchart outlining the methodology for literature search.**

typically available only in full-text articles. Therefore, we selected full-text, peer-reviewed articles to ensure methodological rigor and data reliability. Randomized controlled trials (RCTs) were excluded from our eligibility criteria because their primary focus on interventions and selective participant groups often makes baseline data less representative and insufficiently detailed for accurate prevalence estimation. Similarly, letters to the editor and other short communications were excluded as they generally lack critical methodological information, including study design, sample size, and population characteristics. Furthermore, studies published before 2000 were omitted to ensure the findings reflect current

epidemiological conditions, considering major advancements in healthcare, vaccination, hygiene, and surveillance across South Asia over the past two decades. Finally, the term 'healthy population' used in this study refers to individuals who do not have any serious health issues that could influence immune function or viral susceptibility. This definition includes people with minor or self-limiting illnesses (e.g., common cold) but excludes those with chronic medical conditions (such as cancer, HIV, diabetes, cardiovascular disease and others), immunocompromised status, or conditions requiring long-term medical care. As per our study objective, we focus on individuals who are not affected by underlying health conditions that might skew the prevalence of viral infections. Based on these considerations, the following inclusion and exclusion criteria were applied.

Inclusion criteria: 1. Studies published in English, with full-text availability, and reporting the prevalence of any of these viruses CMV, EBV, HPV, and HSV; 2. Original, peer-reviewed observational studies [cross-sectional, retrospective, prospective, pilot, and case-control studies (using only the control groups)]; 3. Healthy individuals (with minor illnesses unrelated to chronic diseases); 4. Conducted in South Asian countries (Afghanistan, Bangladesh, Bhutan, India, Maldives, Nepal, Pakistan, and Sri Lanka); 5. Articles published from 2000 to 2025.

Exclusion criteria: 1. Secondary sources (Review articles, editorials, case reports); 2. Individuals with chronic medical conditions (Cancer, chronic kidney disease, chronic heart disease, HIV) or group of population (pregnant women, and sex workers); 3. Studies that did not report the prevalence of viruses; 4. Studies that met the inclusion criteria but the full text could not be retrieved from the authors after requests.

The Mendeley Desktop software (version 1.19.4) was used to organize the references and remove the duplicates. Studies were independently verified by two co-authors (AKS and BB) before final inclusion in the meta-analysis. Disagreements were resolved through group discussions between all co-authors.

## Data extraction

The eligible studies were divided among the three co-authors (RTS, ZAS, and NMN) who independently used an Excel table to systematically extract information. The extracted data were subsequently cross-checked by another co-author (BB) to ensure accuracy and consistency. The information that was extracted for each study were: publication details [e.g., first author, publication date]; study setting (community, population, or hospital-based); population and study design [e.g., country, study location, study period, and sample size]; ethical statements and consent (written, oral, or verbal), characteristics of participants [e.g., gender, age] and key findings (e.g., testing methods, sample type, and viral prevalence). The accuracy of the extracted data was verified by all co-authors through multiple revisions of the included studies. In cases where viral prevalence was reported as a percentage, conversions to decimal format were performed to ensure consistency across the dataset.

## Evaluation of study quality

Study quality was evaluated by following the 9-item tool explicitly developed by Joanna Briggs Institute for prevalence study [48]. Based on the critical appraisal tool scoring system, if the answer was "Yes", then 1 was implied and if the answer was "No" or "Unclear" then 0 was implied. Each study was categorized based on its total score as either "low-risk bias" (8–9), "moderate risk bias" (6–7), or "high-risk bias" (0–5). Quality assessment was conducted for all selected articles independently by two co-authors (RTS, ZAS). Disagreements among co-authors (RTS, ZAS) during the assessment were resolved through discussions with the project supervisor (AKS) to uphold consistency.

## Statistical analysis

The prevalence of CMV, EBV, HPV, and HSV were considered as summary measurements. We used a DerSimonian-Laird random-effect model to obtain pooled prevalence with a 95% confidence interval. Country-wise and gender-wise analysis was also performed according to the prevalence of the participants. Heterogeneity was assessed

 

using Cochran's Q test and the $I^2$ statistics. Substantial heterogeneity was indicated with an $I^2$ of more than 75% [49]. Publication bias was also examined using a funnel plot. All statistical analyses were performed using Microsoft Excel version 16.

## Result

We found 6,307 studies using the search strategy mentioned earlier. Among these, 6,223 articles were excluded, and finally, 94 studies comprising 162,659 healthy individuals were included in this systematic review and meta-analysis [50–143]. The details of the study articles selection process are shown in Fig 1.

Out of 94 studies, 67 were conducted in India with a population of 134,079. Among the rest, 8 were conducted in Pakistan (3,295 participants), 5 in Nepal (3,711 participants), 4 in Bangladesh (4,075 individuals), 4 in Sri Lanka (2,506 individuals), 2 in Bhutan (5,137 individuals), and 2 in Afghanistan (5,235 individuals). We found that more than half of the included studies were designed in a hospital-based setting (49 out of 94), and the rest were community-based (44 out of 94). Additionally, 1 study did not specify the study setting. More than one-third of the studies did not mention their study design (35 out of 94) [59,67–74,76,79,82,86,89,90,93,96,97,99–101,107,108,111,114–116,119,122–124,126,127,137] and the remaining studies consisted of cross-sectional (46 out of 94) [50,52–57,60,61,64–66,75,77,78,80,81,83–85,87,88,91,94,103–105,109,117,118,125,128–136,138–143], case-control (11 out of 94) [51,58,95,98,102,106,110,112,113,120,121], and cohort (3 out of 94) [62,63,92]. To detect the presence of CMV, EBV, HPV, and HSV the studies used several kinds of testing methods. Among these polymerase chain reaction (PCR) (46 out of 94) and enzyme-linked immunosorbent assay (ELISA) (31 out of 94) were mostly used, only 3 studies mentioned the use of both PCR and ELISA. Other methods included Hybrid-capture 2 (HC2) (6 out of 94), APTIMA HPV (2 out of 94), CareHPV Test (2 out of 94), immunohistochemistry (IHC) (2 out of 94), and chemiluminescence immunoassay (CLIA) (1 out of 94) and Southern blot (1 out of 94). Roughly two-third (43 out of 94) studies used blood samples for detecting viral infection, while other studies utilized cervical specimens (41 out of 94), urine (5 out of 94), tissue (2 out of 94), oral specimens (2 out of 94) or normal mucosa (1 out of 94). The characteristics of the selected studies are presented in Table 1 and S2 Table.

The overall pooled prevalence of these viruses was 20% [95% CI: 16% to 24%], with high degree of heterogeneity ($I^2 = 100\%$; $p < 0.01$). The overall prevalence is shown in the Forest plot (Fig 2).

### Analysis according to virus

This study included the prevalence of four viruses (CMV, EBV, HPV, and HSV) among healthy populations in seven South Asian countries: Afghanistan, Bangladesh, Bhutan, India, Pakistan, Nepal, and Sri Lanka. No studies were found from the Maldives regarding the prevalence of any of these viruses.

**CMV.** We retrieved 17 studies, with a total of 12,453 healthy individuals reporting the prevalence of CMV infection. Country-wise analysis showed that most of the studies were conducted in India (15 out of 17) with a pooled prevalence of 48% [95% CI: 12% to 85%] (S1 Fig). Single studies were conducted in Afghanistan and Pakistan (prevalence ranging from 91% to 100%). All these studies were either hospital-based (11 out of 17) or community-based (5 out of 17), with 1 study did not mention the study setting. The study designs were either cross-sectional (5 out of 17), or case-control (2 out of 17). Most of the studies did not mention study design (10 out of 17). A larger number of studies did not have gender-specific prevalence, while 2 studies were conducted only in males [132,133]. The pooled prevalence of CMV was 57% [95% CI: 21% to 89%] with a significant amount of heterogeneity ($I^2 = 100\%$; $p < 0.01$) (Fig 3).

**EBV.** We found 18 studies involving 3,914 healthy individuals reporting on the prevalence of EBV infection. Among them, 15 studies were carried out in India with a prevalence of 22% [95% CI: 8%-40%] (S2 Fig), and the remaining 3 studies were conducted in Pakistan (prevalence ranging from 0% to 16%). Most of the studies were hospital-based (16 out of 18), while 2 were community-based. The study designs varied, including cross-sectional (5 out of 18), case-control

**Table 1. Characteristics of the selected studies.**

| Reference | Country | Study Design | Virus Name | Testing Method | Sample Size | Total Number of Infected Individuals | | | |
|---|---|---|---|---|---|---|---|---|---|
| | | | | | | CMV | EBV | HPV | HSV |
| Schneider et al – 2010 [50] | India | Cross-sectional | HSV | ELISA | 12617 | | | | 947 |
| Ray et al – 2008 [51] | India | Case-control | HSV | ELISA | 801 | | | | 50 |
| Sgaier et al – 2015 [52] | India | Cross-sectional | HSV | ELISA | 1848 | | | | 187 |
| Dave et al – 2012 [53] | India | Cross-sectional | HSV | ELISA | 841 | | | | 29 |
| Ghosh et al – 2019 [54] | India | Cross-sectional | HSV, HPV, CMV, EBV | PCR | 2240 | 1417 | 1444 | 620 | 21 |
| Adamson et al – 2011 [55] | India | Cross-sectional | HSV | ELISA | 897 | | | | 103 |
| Panchanadeswaran et al – 2006 [56] | India | Cross-sectional | HSV | ELISA | 1620 | | | | 214 |
| Schensul et al – 2007 [57] | India | Cross-sectional | HSV | ELISA | 641 | | | | 62 |
| Patil et al – 2020 [58] | India | Case-control | HSV | ELISA | 125 | | | | 10 |
| Banandur et al – 2011 [59] | India | Not given | HSV | ELISA | 12180 | | | | 1413 |
| Mir et al – 2009 [60] | Pakistan | Cross-sectional | HSV | ELISA | 2383 | | | | 83 |
| Johnson et al – 2014 [61] | Nepal | Cross-sectional | HPV | APTIMA HPV assay | 261 | | | 25 | |
| Sharmin et al – 2021 [62] | Bangladesh | Cohort | HPV | PCR | 410 | | | 121 | |
| Nahar et al – 2014 [63] | Bangladesh | Cohort | HPV | PCR | 1902 | | | 145 | |
| Parwez et al – 2022 [64] | India | Cross-sectional | HPV | PCR | 154 | | | 19 | |
| Sureshkumar et al – 2015 [65] | India | Cross-sectional | HPV | PCR | 1699 | | | 179 | |
| Peedicayil et al – 2009 [66] | India | Cross-sectional | HPV | PCR | 58 | | | 13 | |
| Datta et al – 2010 [67] | India | Not given | HPV | PCR | 1300 | | | 145 | |
| Dutta et al – 2012 [68] | India | Not given | HPV | PCR | 2313 | | | 212 | |
| Sherpa et al – 2009 [69] | Nepal | Not given | HPV | PCR | 898 | | | 73 | |
| Thilagavathi et al – 2012 [70] | India | Not given | HPV | PCR | 238 | | | 22 | |
| Franceschi et al – 2005 [71] | India | Not given | HPV | PCR | 1891 | | | 252 | |
| Mittal et al – 2015 [72] | India | Not given | HPV | HC2 | 43313 | | | 2055 | |
| Silver et al – 2011 [73] | India | Not given | HPV, EBV, CMV | PCR | 464 | 121 | 93 | 93 | |
| Hussain et al – 2012 [74] | India | Not given | HPV | PCR | 800 | | | 22 | |
| Johnson et al – 2016 [75] | Nepal | Cross-sectional | HPV | APTIMA HPV assay | 265 | | | 20 | |
| Aziz et al – 2023 [76] | Pakistan | Not given | HPV | PCR | 135 | | | 6 | |
| Shahid et al – 2015 [77] | Pakistan | Cross-sectional | HPV | PCR | 160 | | | 17 | |
| Baussano et al – 2017 [78] | Bhutan | Cross-sectional | HPV | careHPV test | 2590 | | | 265 | |
| Becker et al – 2007 [79] | India | Not given | HSV | ELISA | 901 | | | | 170 |
| Parvez et al – 2023 [80] | India | Cross-sectional | HPV | PCR | 1000 | | | 36 | |
| Clifford et al – 2023 [81] | Bhutan | Cross-sectional | HPV | careHPV test | 2547 | | | 260 | |
| Shakya et al – 2018 [82] | Nepal | Not given | HPV | PCR | 1289 | | | 186 | |
| Todd et al – 2012 [83] | Afghanistan | Cross-sectional | HSV | ELISA | 4750 | | | | 144 |
| Ramesh et al – 2021 [84] | India | Cross-sectional | HPV | PCR | 200 | | | 0 | |
| Subramanian et al – 2021 [85] | India | Cross-sectional | HPV | HC2 | 1523 | | | 193 | |
| Dakshinamurthy et al – 2023 [86] | India | Not given | HPV | PCR | 1035 | | | 270 | |
| Mishra et al – 2022 [87] | India | Cross-sectional | HPV | PCR | 217 | | | 12 | |
| Shashidhar et al – 2021 [88] | India | Cross-sectional | HPV | PCR | 126 | | | 9 | |
| Bhattacharya et al – 2018 [89] | India | Not given | HPV | PCR | 684 | | | 252 | |
| Asiaf et al – 2012 [90] | India | Not given | HPV | PCR | 210 | | | 29 | |
| Sauvaget et al – 2011 [91] | India | Cross-sectional | HPV | HC2 | 27192 | | | 2812 | |
| Sharma et al – 2015 [92] | India | Cohort | HPV | PCR | 2034 | | | 262 | |
| Srivastava et al – 2012 [93] | India | Not given | HPV | PCR | 2424 | | | 240 | |

*(Continued)*

| Reference | Country | Study Design | Virus Name | Testing Method | Sample Size | Total Number of Infected Individuals | | | |
|---|---|---|---|---|---|---|---|---|---|
| | | | | | | CMV | EBV | HPV | HSV |
| Mapitigama et al – 2023 [94] | Sri Lanka | Cross-sectional | HPV | PCR | 1012 | | | 56 | |
| Vinodhini et al – 2012 [95] | India | Case-control | HPV | PCR | 257 | | | 78 | |
| Khanna et al – 2009 [96] | India | Not given | HPV | Southern blot | 45 | | | 9 | |
| Naushad et al – 2017 [97] | Pakistan | Not given | HPV, EBV | PCR | 100 | | 0 | 0 | |
| Gunasekera et al – 2015 [98] | Sri Lanka | Case-control | HPV | ELISA | 51 | | | 7 | |
| Saranath et al – 2001 [99] | India | Not given | HPV | PCR | 164 | | | 62 | |
| Gopalkrishna et al – 2000 [100] | India | Not given | HPV | PCR | 30 | | | 4 | |
| Pandit et al – 2013 [101] | India | Not given | EBV | ELISA | 140 | | 138 | | |
| Lourembam et al – 2015 [102] | India | Case-control | EBV | PCR | 115 | | 24 | | |
| Janani et al – 2015 [103] | India | Cross-sectional | EBV, HSV, CMV | PCR | 60 | | 8 | | |
| Janani et al – 2015 [104] | India | Cross-sectional | EBV, CMV | ELISA, PCR | 60 | | 35 | | |
| Sinha et al – 2015 [105] | India | Cross-sectional | EBV | PCR | 30 | | 0 | | |
| Ghosh et al – 2014 [106] | India | Case-control | EBV | PCR | 100 | | 54 | | |
| Noorali et al – 2004 [107] | Pakistan | Not given | EBV | PCR | 30 | | 0 | | |
| Borthakur et al – 2016 [108] | India | Not given | EBV | IHC | 20 | | 0 | | |
| Chatterjee et al – 2022 [109] | India | Cross-sectional | EBV | PCR | 130 | | 14 | | |
| Sangam et al – 2019 [110] | India | Case-control | EBV | PCR | 50 | | 8 | | |
| Sachithanandham et al – 2013 [111] | India | Not given | EBV, CMV | PCR | 70 | | 9 | | 0 |
| Reddy et al – 2016 [112] | India | Case-control | EBV | IHC | 25 | | 2 | | |
| Sharma et al – 2019 [113] | India | Case-control | EBV, HPV | PCR | 150 | | 14 | 11 | |
| Rizvi et al – 2011 [114] | India | Not given | EBV, CMV | ELISA | 30 | 1 | 0 | | |
| Husseini et al – 2019 [115] | Afghanistan | Not given | CMV | CLIA | 485 | 484 | | | |
| Chakravarti et al – 2010 [116] | India | Not given | CMV | ELISA | 60 | 3 | | | |
| Das et al – 2014 [117] | India | Cross-sectional | CMV | ELISA | 2100 | 2070 | | | |
| Chaudhari et al – 2009 [118] | India | Cross-sectional | CMV | ELISA | 431 | 379 | | | |
| Surpam et al – 2005 [119] | India | Not given | CMV, HSV | ELISA | 75 | 1 | | | 3 |
| Tewari et al – 2011 [120] | India | Case-control | CMV | ELISA | 200 | 130 | | | |
| Dubey et al – 2020 [121] | India | Case-control | CMV | ELISA, PCR | 42 | 40 | | | |
| Anuradha et al – 2011 [122] | India | Not given | CMV | ELISA | 52 | 6 | | | |
| Mujtaba et al – 2001 [123] | India | Not given | CMV | ELISA, PCR | 32 | 1 | | | |
| Kothari et al – 2002 [124] | India | Not given | CMV | ELISA | 200 | 190 | | | |
| Kumar et al – 2008 [125] | India | Cross-sectional | CMV | ELISA | 5600 | 4 | | | |
| Sharma et al – 2007 [126] | India | Not given | CMV | ELISA | 25 | 25 | | | |
| Padmavati et al – 2012 [127] | India | Not given | CMV | ELISA | 200 | 159 | | | |
| Thapa et al – 2018 [128] | Nepal | Cross-sectional | HPV | PCR | 998 | | | 197 | |
| Perera et al – 2021 [129] | Sri Lanka | Cross-sectional | HPV | PCR | 822 | | | 51 | |
| Gibney et al – 2001 [130] | Bangladesh | Cross-sectional | HSV | ELISA | 387 | | | | 100 |
| Ibrahim et al – 2016 [131] | Pakistan | Cross-sectional | CMV | ELISA | 217 | 205 | | | |
| Hawkes et al – 2002 [132] | Bangladesh | Cross-sectional | HSV | ELISA | 312 | | | | 18 |
| Munir et al – 2023 [133] | Pakistan | Cross-sectional | EBV | PCR | 100 | | 9 | | |
| Perera et al – 2024 [134] | Sri Lanka | Cross-sectional | HPV | PCR | 621 | | | 38 | |
| Minhas et al – 2024 [135] | Pakistan | Cross-sectional | HPV | PCR | 170 | | | 83 | |
| Mittal et al – 2024 [136] | India | Cross-sectional | HPV | PCR | 653 | | | 32 | |
| Oommen et al -2024 [137] | India | Not given | HPV | PCR | 1170 | | | 78 | |

*(Continued)*

 

**Table 1.** (Continued)

| Reference | Country | Study Design | Virus Name | Testing Method | Sample Size | Total Number of Infected Individuals | | | |
|---|---|---|---|---|---|---|---|---|---|
| | | | | | | CMV | EBV | HPV | HSV |
| Panta et al- 2024 [138] | India | Cross-sectional | HPV | HC2 | 975 | | | 45 | |
| Parvez et al – 2024 [139] | India | Cross-sectional | HPV | PCR | 1000 | | | 59 | |
| Munni et al – 2024 [140] | India | Cross-sectional | HPV | HC2 | 501 | | | 32 | |
| Chakroborty et al – 2024 [141] | Bangladesh | Cross-sectional | HPV | PCR | 900 | | | 23 | |
| Khoja et al – 2024 [142] | Bangladesh | Cross-sectional | HPV | HC2 | 164 | | | 7 | |
| Deka et al -2024 [143] | India | Cross-sectional | HSV | ELISA | 322 | | | | 148 |

Abbreviations: *ELISA* Enzyme-linked Immunosorbent Assay, *PCR* Polymerase Chain Reaction, *HC2* Hybrid-Capture 2, *IHC* Immunohistochemistry, *CLIA* Chemiluminescence Immunoassay.

(6 out of 18), and not mentioned (7 out of 18). The majority of the studies included both male and female participants, while 3 studies were only conducted on males. The overall prevalence of EBV was 17% [95% CI: 5% to 34%] with a high degree of heterogeneity ($I^2 = 99\%$; $p < 0.01$) (Fig 4).

**HPV.** We found 51 studies comprising 111,355 healthy individuals that reported HPV infection. The majority of these studies were conducted in India (32 out of 51), revealing a pooled prevalence rate of 12% [95% CI: 9% to 14%] (S3 Fig). Furthermore, we found 5 studies from Nepal (prevalence ranging from 5% to 22%), 2 from Bhutan (prevalence ranging from 9% to 11%), 4 from Bangladesh (prevalence ranging from 4.2% to 29.5%), 4 from Pakistan (prevalence ranging from 0% to 48.8%), and 4 from Sri Lanka (prevalence ranging from 5.5% to 99.8%). The study settings of these studies were either hospital-based (21 out of 51) or community-based (30 out of 51). While most of the studies were cross-sectional (26 out of 51), the remaining were case-control (3 out of 51), cohort (3 out of 51) or not mentioned (19 out of 51). Out of 51 studies, 46 studies were conducted only on female participants with a prevalence rate of 13% [95% CI: 10% to 16%] (S4 Fig) while the rest included both males and females. The overall prevalence of HPV infection was 13% [95% CI: 10% to 16%] with a significant heterogeneity ($I^2 = 99\%$; $p < 0.01$) (Fig 5).

**HSV.** We found 18 studies reporting the prevalence of HSV infection, involving a total of 43,010 healthy individuals. Among these, 14 studies were conducted in India with a pooled prevalence of 9% [95% CI: 6% to 12%] (S5 Fig), while 2 were conducted in Bangladesh (prevalence ranging from 3% to 30%), and single studies were conducted in Afghanistan and Pakistan (prevalence ranging from 3% to 4%). The majority of the study settings were either hospital-based (4 out of 18) or community-based (13 out of 18) and 1 study did not specify the study setting. The study designs included cross-sectional (13 out of 18), case-control (2 out of 18) and not mentioned (3 out of 18). Out of these studies, 5 focused solely on male participants with a prevalence of 10% [95% CI: 6% to 14%] (S6 Fig), and 5 studies on females with a prevalence rate of 10% [95% CI: 5% to 17%] (S7 Fig). The overall prevalence of HSV infection was 9% [95% CI: 6% to 12%] with a significant amount of heterogeneity ($I^2 = 100\%$; $p < 0.01$) (Fig 6).

## Quality assessment and publication bias

Critical Appraisal Tool for prevalence study developed by the Joanna Briggs Institute was used for the evaluation of selected study articles. Out of 94 selected articles, 67 showed a low risk of bias, 25 had a moderate risk of bias, and only 2 were found to contain a high risk of bias (S3 Table). Additionally, funnel plots indicating the existence of asymmetry and publication bias for the overall prevalence of these viruses (HSV, HPV, CMV, and EBV) are presented in a supporting file; S8 Fig.

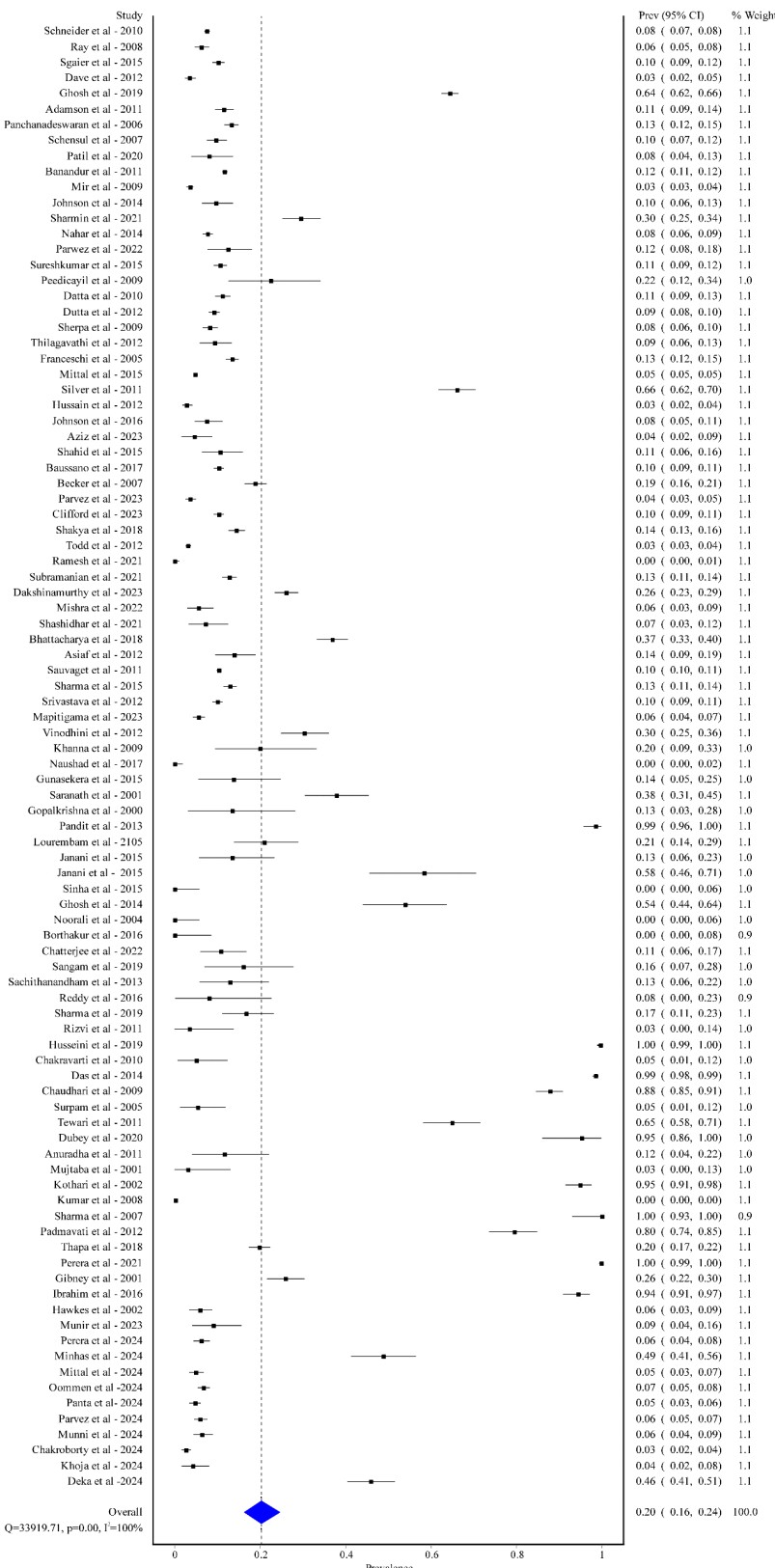

| Study | Prev (95% CI) | % Weight |
|---|---|---|
| Schneider et al - 2010 | 0.08 ( 0.07, 0.08) | 1.1 |
| Ray et al - 2008 | 0.06 ( 0.05, 0.08) | 1.1 |
| Sgaier et al - 2015 | 0.10 ( 0.09, 0.12) | 1.1 |
| Dave et al - 2012 | 0.03 ( 0.02, 0.05) | 1.1 |
| Ghosh et al - 2019 | 0.64 ( 0.62, 0.66) | 1.1 |
| Adamson et al - 2011 | 0.11 ( 0.09, 0.14) | 1.1 |
| Panchanadeswaran et al - 2006 | 0.13 ( 0.12, 0.15) | 1.1 |
| Schensul et al - 2007 | 0.10 ( 0.07, 0.12) | 1.1 |
| Patil et al - 2020 | 0.08 ( 0.04, 0.13) | 1.1 |
| Banandur et al - 2011 | 0.12 ( 0.11, 0.12) | 1.1 |
| Mir et al - 2009 | 0.03 ( 0.03, 0.04) | 1.1 |
| Johnson et al - 2014 | 0.10 ( 0.06, 0.13) | 1.1 |
| Sharmin et al - 2021 | 0.30 ( 0.25, 0.34) | 1.1 |
| Nahar et al - 2014 | 0.08 ( 0.06, 0.09) | 1.1 |
| Parwez et al - 2022 | 0.12 ( 0.08, 0.18) | 1.1 |
| Sureshkumar et al - 2015 | 0.11 ( 0.09, 0.12) | 1.1 |
| Peedicayil et al - 2009 | 0.22 ( 0.12, 0.34) | 1.0 |
| Datta et al - 2010 | 0.11 ( 0.09, 0.13) | 1.1 |
| Dutta et al - 2012 | 0.09 ( 0.08, 0.10) | 1.1 |
| Sherpa et al - 2009 | 0.08 ( 0.06, 0.10) | 1.1 |
| Thilagavathi et al - 2012 | 0.09 ( 0.06, 0.13) | 1.1 |
| Franceschi et al - 2005 | 0.13 ( 0.12, 0.15) | 1.1 |
| Mittal et al - 2015 | 0.05 ( 0.05, 0.05) | 1.1 |
| Silver et al - 2011 | 0.66 ( 0.62, 0.70) | 1.1 |
| Hussain et al - 2012 | 0.03 ( 0.02, 0.04) | 1.1 |
| Johnson et al - 2016 | 0.08 ( 0.05, 0.11) | 1.1 |
| Aziz et al - 2023 | 0.04 ( 0.02, 0.09) | 1.1 |
| Shahid et al - 2015 | 0.11 ( 0.06, 0.16) | 1.1 |
| Baussano et al - 2017 | 0.10 ( 0.09, 0.11) | 1.1 |
| Becker et al - 2007 | 0.19 ( 0.16, 0.21) | 1.1 |
| Parvez et al - 2023 | 0.04 ( 0.03, 0.05) | 1.1 |
| Clifford et al - 2023 | 0.10 ( 0.09, 0.11) | 1.1 |
| Shakya et al - 2018 | 0.14 ( 0.13, 0.16) | 1.1 |
| Todd et al - 2012 | 0.03 ( 0.03, 0.04) | 1.1 |
| Ramesh et al - 2021 | 0.00 ( 0.00, 0.01) | 1.1 |
| Subramanian et al - 2021 | 0.13 ( 0.11, 0.14) | 1.1 |
| Dakshinamurthy et al - 2023 | 0.26 ( 0.23, 0.29) | 1.1 |
| Mishra et al - 2022 | 0.06 ( 0.03, 0.09) | 1.1 |
| Shashidhar et al - 2021 | 0.07 ( 0.03, 0.12) | 1.1 |
| Bhattacharya et al - 2018 | 0.37 ( 0.33, 0.40) | 1.1 |
| Asiaf et al - 2012 | 0.14 ( 0.09, 0.19) | 1.1 |
| Sauvaget et al - 2011 | 0.10 ( 0.10, 0.11) | 1.1 |
| Sharma et al - 2015 | 0.13 ( 0.11, 0.14) | 1.1 |
| Srivastava et al - 2012 | 0.10 ( 0.09, 0.11) | 1.1 |
| Mapitigama et al - 2023 | 0.06 ( 0.04, 0.07) | 1.1 |
| Vinodhini et al - 2012 | 0.30 ( 0.25, 0.36) | 1.1 |
| Khanna et al - 2009 | 0.20 ( 0.09, 0.33) | 1.0 |
| Naushad et al - 2017 | 0.00 ( 0.00, 0.02) | 1.1 |
| Gunasekera et al - 2015 | 0.14 ( 0.05, 0.25) | 1.0 |
| Saranath et al - 2001 | 0.38 ( 0.31, 0.45) | 1.1 |
| Gopalkrishna et al -2000 | 0.13 ( 0.03, 0.28) | 1.0 |
| Pandit et al - 2013 | 0.99 ( 0.96, 1.00) | 1.1 |
| Lourembam et al - 2105 | 0.21 ( 0.14, 0.29) | 1.1 |
| Janani et al - 2015 | 0.13 ( 0.06, 0.23) | 1.0 |
| Janani et al - 2015 | 0.58 ( 0.46, 0.71) | 1.0 |
| Sinha et al - 2015 | 0.00 ( 0.00, 0.06) | 1.0 |
| Ghosh et al - 2014 | 0.54 ( 0.44, 0.64) | 1.1 |
| Noorali et al - 2004 | 0.00 ( 0.00, 0.06) | 1.0 |
| Borthakur et al - 2016 | 0.00 ( 0.00, 0.08) | 0.9 |
| Chatterjee et al - 2022 | 0.11 ( 0.06, 0.17) | 1.1 |
| Sangam et al - 2019 | 0.16 ( 0.07, 0.28) | 1.0 |
| Sachithanandham et al - 2013 | 0.13 ( 0.06, 0.22) | 1.0 |
| Reddy et al - 2016 | 0.08 ( 0.00, 0.23) | 0.9 |
| Sharma et al - 2019 | 0.17 ( 0.11, 0.23) | 1.1 |
| Rizvi et al - 2011 | 0.03 ( 0.00, 0.14) | 1.0 |
| Husseini et al - 2019 | 1.00 ( 0.99, 1.00) | 1.1 |
| Chakravarti et al - 2010 | 0.05 ( 0.01, 0.12) | 1.0 |
| Das et al - 2014 | 0.99 ( 0.98, 0.99) | 1.1 |
| Chaudhari et al - 2009 | 0.88 ( 0.85, 0.91) | 1.1 |
| Surpam et al - 2005 | 0.05 ( 0.01, 0.12) | 1.0 |
| Tewari et al - 2011 | 0.65 ( 0.58, 0.71) | 1.1 |
| Dubey et al - 2020 | 0.95 ( 0.86, 1.00) | 1.0 |
| Anuradha et al - 2011 | 0.12 ( 0.04, 0.22) | 1.0 |
| Mujtaba et al - 2001 | 0.03 ( 0.00, 0.13) | 1.0 |
| Kothari et al - 2002 | 0.95 ( 0.91, 0.98) | 1.1 |
| Kumar et al - 2008 | 0.00 ( 0.00, 0.00) | 1.1 |
| Sharma et al - 2007 | 1.00 ( 0.93, 1.00) | 0.9 |
| Padmavati et al - 2012 | 0.80 ( 0.74, 0.85) | 1.1 |
| Thapa et al - 2018 | 0.20 ( 0.17, 0.22) | 1.1 |
| Perera et al - 2021 | 1.00 ( 0.99, 1.00) | 1.1 |
| Gibney et al - 2001 | 0.26 ( 0.22, 0.30) | 1.1 |
| Ibrahim et al - 2016 | 0.94 ( 0.91, 0.97) | 1.1 |
| Hawkes et al - 2002 | 0.06 ( 0.03, 0.09) | 1.1 |
| Munir et al - 2023 | 0.09 ( 0.04, 0.16) | 1.1 |
| Perera et al - 2024 | 0.06 ( 0.04, 0.08) | 1.1 |
| Minhas et al - 2024 | 0.49 ( 0.41, 0.56) | 1.1 |
| Mittal et al - 2024 | 0.05 ( 0.03, 0.07) | 1.1 |
| Oommen et al -2024 | 0.07 ( 0.05, 0.08) | 1.1 |
| Panta et al- 2024 | 0.05 ( 0.03, 0.06) | 1.1 |
| Parvez et al - 2024 | 0.06 ( 0.05, 0.07) | 1.1 |
| Munni et al - 2024 | 0.06 ( 0.04, 0.09) | 1.1 |
| Chakroborty et al - 2024 | 0.03 ( 0.02, 0.04) | 1.1 |
| Khoja et al - 2024 | 0.04 ( 0.02, 0.08) | 1.1 |
| Deka et al -2024 | 0.46 ( 0.41, 0.51) | 1.1 |
| Overall | 0.20 ( 0.16, 0.24) | 100.0 |

Q=33919.71, p=0.00, I²=100%

Prevalence

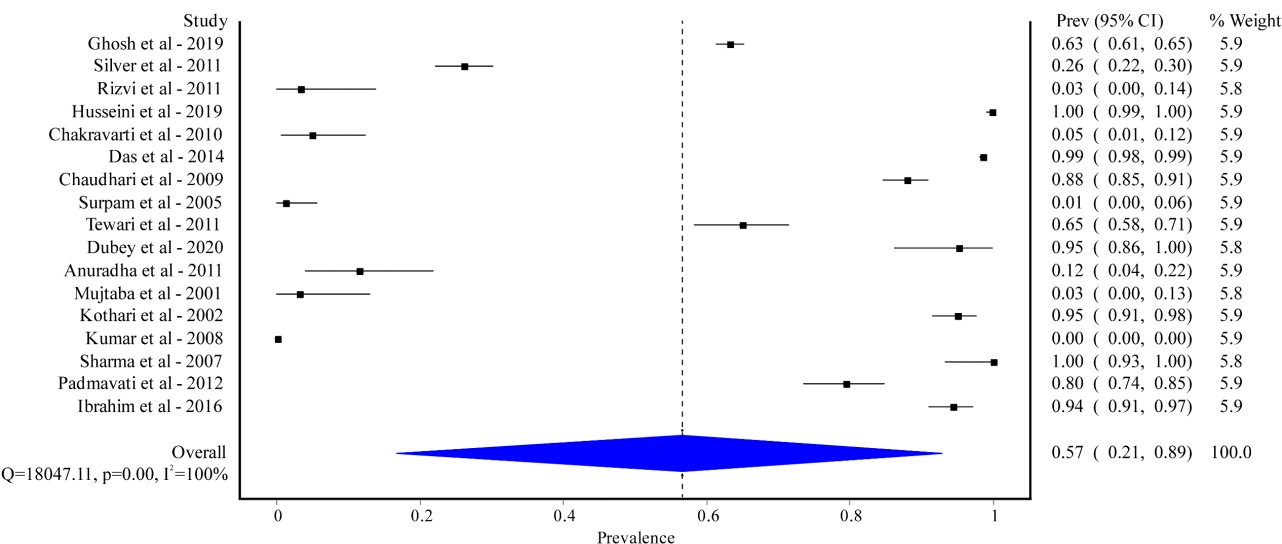

**Fig 2. Overall pooled prevalence of CMV, EBV, HPV, and HSV among healthy populations in South Asian.**

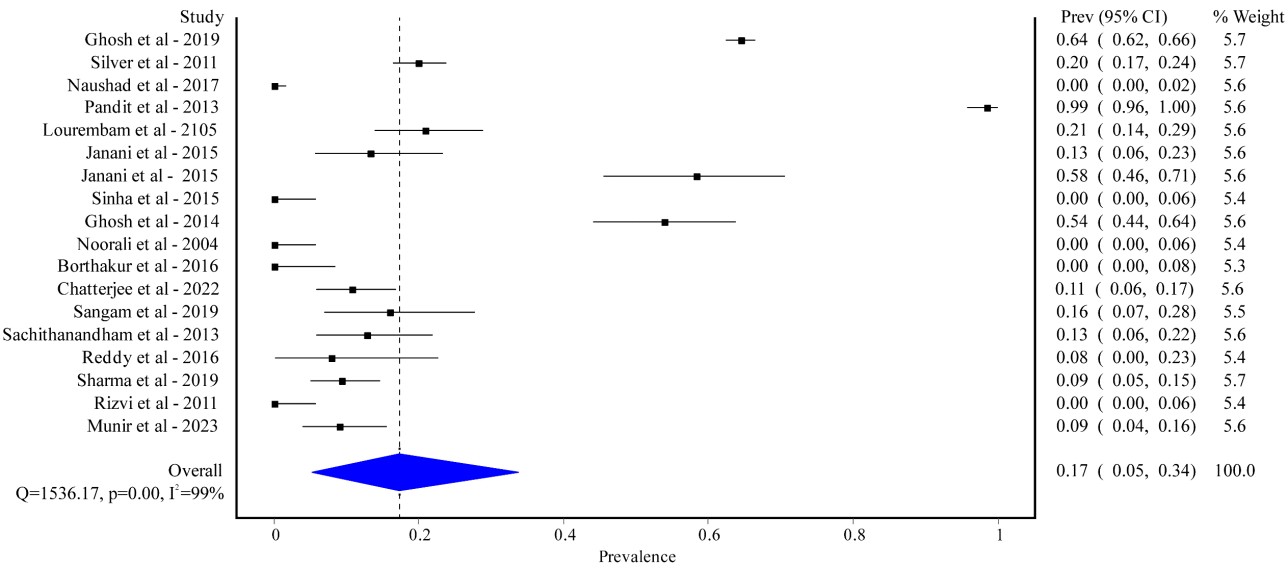

**Fig 3. Pooled prevalence of CMV among healthy populations in South Asian.**

**Fig 4. Pooled prevalence of EBV among healthy populations in South Asian.**

## Discussion

Our findings highlighted that the overall pooled prevalence of these viruses is 20% [95% CI: 16% to 24%]. Among these four viruses, CMV infection had the highest prevalence 57% [95% CI: 21% to 89%] and HSV had the lowest 9% [95% CI: 6% to 12%]. This result indicates that compared to the other viruses, CMV transmission is more widespread in South Asia,

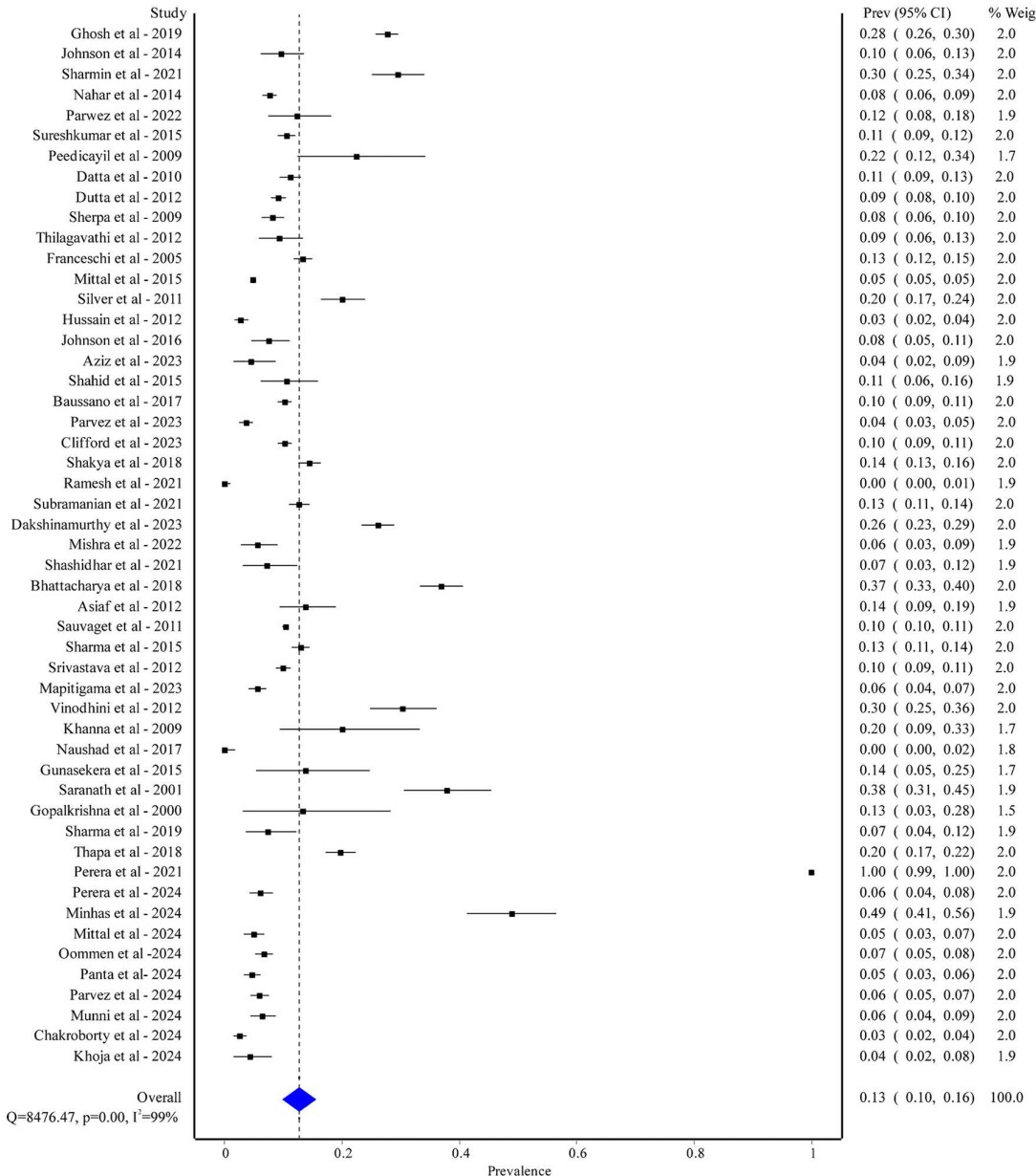

**Fig 5. Pooled prevalence of HPV among healthy populations in South Asian.**

highlighting the need for targeted interventions and further research into the factors contributing to its high prevalence. Our result also pointed out that 2 out of 10 healthy individual residing in South Asian countries is likely to encounter one of these viruses and develop a lifelong infection due to their latent infectious capabilities. Therefore, a comprehensive approach is essential to accurately determine the burden of these viruses in South Asia and design targeted public health strategies.

A study conducted in Poland reported the prevalence of the same four viruses and found that 30% of healthy individuals were infected with at least one of them [144]. The relatively lower prevalence in our study could be attributed to the diverse sample sizes across South Asian countries, in contrast to the smaller and more homogeneous samples

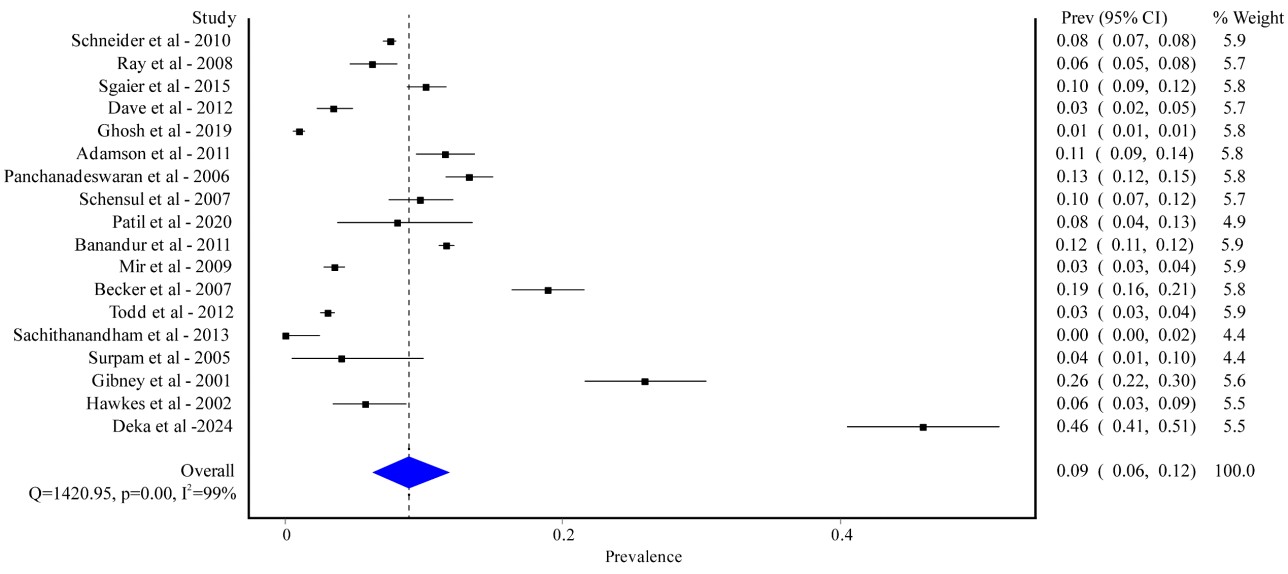

**Fig 6. Pooled prevalence of HSV among healthy populations in South Asian.**

from Poland. Additionally, our study showed that in India the prevalence of CMV, EBV, HPV, and HSV was 48%, 22%, 12%, and 9%, respectively. When compared with similar studies across the globe, CMV shows a prevalence of 86% in the WHO Southeast Asian region among the general population and EBV shows a prevalence of 95.4% among healthy adults in China [145,146]. Our estimate for HPV aligns more closely with Northern American women with normal cytology (11.3%) whereas Europe shows a lower prevalence (8.1%) [147]. In Australia, a nationwide population survey reported a broader range of HSV prevalence (12–76%) compared to our findings [148]. These variations highlight regional differences that might be influenced by factors such as healthcare access, and cultural practices. Studies also support this argument by demonstrating that regional disparities in viral infections can arise from cultural norms that influence vaccination intentions [149], as well as from inequalities in healthcare access, including insurance coverage and regular medical care [150].

A concerning finding from our study was that among healthy females in South Asia, the prevalence of HPV was 13%, which contradicts the 4.4% prevalence reported in individuals with normal cytology in the same region according to the Human Papillomavirus and Related Diseases Report [151]. A plausible explanation might be that healthy females in the South Asian general population might not have undergone regular screening for HPV. As a result, a higher prevalence was observed when they were eventually tested. Another gender-specific finding from our study was that HSV prevalence was slightly higher in women (11%) than in men (10%), which contrasts with the reported fact that HSV infection is typically twice as common in women compared to men [8]. This difference could be attributed to socio-cultural factors, such as socio-economic conditions and cultural behaviors within specific racial or educational groups [152–154], influencing the varying rates of HSV infection [155].

Although to the best of our knowledge, this is the first systematic review and meta-analysis that combined determines the prevalence of CMV, EBV, HPV, and HSV in the South Asian healthy population, the study also has few limitations. We found that a large number of included studies were conducted in India and there were no studies from Maldives suggesting the need for conducting similar research across other South Asian countries. Also, most of the studies included in this review were female-focused, so we could not perform gender-based meta- analysis for all the viruses. Moreover, data for types of viruses were not available in all the studies which restricted us from doing analysis on virus types.

These factors reflected a lack of sub-group analysis in our systematic review. In addition, our meta-analysis showed substantial heterogeneity ($I^2 = 100\%$; $p < 0.01$) due to a number of factors. Some of our included studies demonstrated 0% [84,97,105,107,108,111,114,125] and 100% [92,115] prevalence rates which is mainly due to variations in sample size, study design, and the specific populations being studied. These variabilities influenced the overall weighting of prevalence rates in our statistical analysis. Lastly, using different methods for detecting the viral infection across different studies could introduce assay bias to our findings. However, this limitation is inherent in all similar studies and was, therefore, unavoidable.

The ability of these viruses to persist and reactivate in the host poses a significant challenge for healthcare, necessitating ongoing research into prevention and treatment strategies. For CMV, prevention of congenital infection involves educating pregnant women about transmission risks and promoting hygienic practices. Currently, there are no established antiviral treatments or vaccines for CMV [156]. However, recent advances include the development of replication-defective virus vaccines, which have shown promising immune responses in phase 1 trials [157]. For EBV, prevention primarily focuses on reducing the risk of transmission through measures such as avoiding intimate contact with individuals shedding the virus, particularly during periods of illness, and promoting good hygiene practices [158]. HPV prevention efforts are largely centered on vaccination, targeting young adolescents to reduce the transmission and subsequent development of HPV-related cancers. Additionally, regular screening and treatment of precancerous lesions in women are crucial components of HPV control programs [159]. In the case of HSV, preventive measures involve educating individuals about transmission risks, encouraging disclosure of infection status to partners, practicing safe sex, and considering suppressive therapy in certain cases [160]. Overall, a comprehensive approach to prevention involves a combination of vaccination, education, behavioral interventions, and public health initiatives aimed at reducing transmission and mitigating the burden of these viral infections on global health.

It is estimated that, on average, every person can be simultaneously infected with 8–12 chronic viral infections, whether caused by DNA or RNA viruses [161]. While EBV and CMV can mimic the common cold, HPV and HSV can cause painless lumps or be completely asymptomatic [162–165]. Recent reports have confirmed that after having infections with CMV, and EBV people can develop Guillain-Barré syndrome (GBS) [166]. Moreover, infections with HPV and HSV have detrimental consequences especially in women as HPV infection can lead to cervical cancer and HSV can be transmitted from pregnant women to their neonates [167,168]. Hence, in our study, the prevalence rates of the four DNA (CMV, EBV, HPV, and HSV) viruses suggest the need for regular medical screening.

In conclusion, our study confirms that 22 out of 100 healthy individuals are infected with one of these DNA viruses, emphasizing the widespread impact across different geographical regions in South Asia. Hence, it is crucial to establish preventive guidelines and spread awareness, as these infections can lead to severe health complications. The findings of our study, while offering insights into the prevalence of four viruses in South Asia, can be implemented in public health strategies globally.

## Supporting information

**S1 Fig. Pooled prevalence of CMV among healthy populations in India.**
(JPG)

**S2 Fig. Pooled prevalence of EBV among healthy populations in India.**
(JPG)

**S3 Fig. Pooled prevalence of HPV among healthy populations in India.**
(JPG)

**S4 Fig. Pooled prevalence of HPV among healthy females in India.**
(JPG)

**S5 Fig. Pooled prevalence of HSV among healthy populations in India.**
(JPG)

**S6 Fig. Pooled prevalence of HSV among healthy males.**
(JPG)

**S7 Fig. Pooled prevalence of HSV among healthy females.**
(JPG)

**S8 Fig. Funnel plot.**
(JPG)

**S1 Table. Searching detail, keywords, and alternative terms used in this study.**
(DOCX)

**S2 Table. Characteristics of the selected studies.**
(DOCX)

**S3 Table. Quality assessment of selected studies (Q1, Q2, Q3……Q9 denotes nine parameters described by Joanna Briggs Institute Prevalence Critical Appraisal Tool).**
(DOCX)

**S1 Checklist. PRISMA checklist.** The PRISMA 2020 checklist is reproduced from Page MJ et al. BMJ 2021 under the Creative Commons Attribution 4.0 International (CC BY 4.0) license. https://doi.org/10.1371/journal.pntd.0013853.s001.
(DOCX)

## Author contributions

**Conceptualization:** Akash Ahmed.

**Data curation:** Rifa Tamanna Subarna, Nafisa Mehreen Naser, Nabila Khan.

**Formal analysis:** Rifa Tamanna Subarna.

**Investigation:** Zwad Al Saiyan, Rifa Tamanna Subarna.

**Methodology:** Zwad Al Saiyan, Rifa Tamanna Subarna, Nabila Khan.

**Project administration:** Akash Ahmed.

**Software:** Zwad Al Saiyan, Badhan Bhattacharjee.

**Supervision:** Akash Ahmed.

**Validation:** Akash Ahmed, Badhan Bhattacharjee.

**Visualization:** Zwad Al Saiyan, Nafisa Mehreen Naser, Badhan Bhattacharjee, Nabila Khan, Nadia Sultana Deen.

**Writing – original draft:** Akash Ahmed.

**Writing – review & editing:** Akash Ahmed, Badhan Bhattacharjee, Nadia Sultana Deen.

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
