## [Decision Letter · Decision Letter 0]

7 May 2025

PGPH-D-24-02429

Prevalence of CMV, EBV, HPV, and HSV among South Asian Healthy Population: A Systematic Review and Meta-Analysis

Dear Mr. Ahmed,

Thank you for submitting your manuscript to PLOS Global Public Health. After careful consideration, we feel that it has merit but does not fully meet PLOS Global Public Health’s publication criteria as it currently stands. Therefore, we invite you to submit a revised version of the manuscript that addresses the points raised during the review process.

We look forward to receiving your revised manuscript.

Kind regards,

Nnodimele Onuigbo Atulomah, PhD

Academic Editor

Journal Requirements:

Additional Editor Comments (if provided):

The manuscript require strengthening which I beleive the reviewers have stated clearly what needs to be rectified. Kindly review the suggestions made by the reviewers and make corrections to make the manuscript meet expected standand.

Reviewers' comments:

Reviewer's Responses to Questions

**Comments to the Author**

1. Does this manuscript meet PLOS Global Public Health’s publication criteria?

Reviewer #1: Partly

Reviewer #2: Yes

2. Has the statistical analysis been performed appropriately and rigorously?

Reviewer #1: No

Reviewer #2: Yes

3. Have the authors made all data underlying the findings in their manuscript fully available (please refer to the Data Availability Statement at the start of the manuscript PDF file)?

Reviewer #1: Yes

Reviewer #2: Yes

4. Is the manuscript presented in an intelligible fashion and written in standard English?

Reviewer #1: Yes

Reviewer #2: Yes

Reviewer #1: Title: Prevalence of CMV, EBV, HPV, and HSV-1 among South Asian Healthy Population: A Systematic Review and Meta-Analysis

Abstract:

• Lines 12-13: Remove the specified sentence as it does not add valuable information and does not fit the context of the abstract.

• Line 30: Revise the unclear and misleading sentence, “Healthy individuals are infected with at least one of the four DNA viruses.” This implies that most or all healthy individuals are infected, which contradicts the overall pooled prevalence reported in the review.

Introduction:

• Lines 59-60: Add a reference to this sentence.

• Line 75: Ensure consistent wording for autism throughout the manuscript. Choose either “autism” or “autism spectrum disorders.”

• Lines 84-87: Adjust the formatting for virus names and prevalence values. Avoid placing virus names in parentheses while simultaneously using parentheses for prevalence values. Maintain consistent formatting to reduce confusion.

• Lines 92-93: Add a reference to this sentence.

• Lines 94-95: Avoid using “Indian” and “Bangladeshi.” Instead, use phrases like “among individuals in India” or “among individuals in Bangladesh”.

Methods:

• Line 111: Add “as shown in S1 Table” to index this table in the text.

• Line 116: Clarify how the three co-authors screened the retrieved records. Specify whether records were divided among them or if each record was screened by at least two reviewers. Provide additional details regarding the title/abstract screening process and the subsequent full-text screening.

• Lines 117-118: Revise this unclear sentence. The statement about conducting a search over four months from November 2023 to December 2024 does not make sense and should be clarified.

• Line 119: Since the search is almost a year old, update it to include the most recent studies.

• Line 127 and S1 Table: Clarify whether MeSH terms were used, as it currently appears that only keywords were utilized.

• Line 129: Avoid repeating the full forms of CDC and WHO as they were already introduced earlier. Use the abbreviations instead.

• Lines 128-130: Provide details about accessing the list of suggested similar articles. Specify whether this was done through CDC and WHO websites. Indicate the number of relevant articles retrieved and include any links, keywords, and the detailed screening methods.

• Line 113: Address whether grey literature was systematically searched.

• Line 133 and Lines 142-144: Provide a justification for restricting the search to full-text articles, as prevalence data and population classifications are frequently reported in abstracts, especially in studies conducted in South Asian populations.

• Lines 134-136: Justify the exclusion of RCTs, considering that these studies can report baseline prevalence data, which is directly relevant to the objectives of this review.

• Line 138: Explain why studies before 2000 were excluded. Provide an epidemiological or methodological justification.

• Line 140: Justify the exclusion of letters to editors, as they may contain relevant primary data on prevalence.

• Line 131: Define “healthy population” as used in this review.

• Line 131: Clarify whether both HSV-1 and HSV-2 were included. Additionally, explain how known issues with serological assays for HSV were addressed in this review.

• Line 150: Include the initials of the three co-authors who extracted data. Describe whether each study was extracted three times or divided among the co-authors to ensure double extraction.

• Line 152: Define what is meant by “population” in the context of study settings.

• Line 168: Specify whether any data transformation was performed to stabilize variance before pooling prevalence data. If not, justify this decision.

• Line 170: Specify the type of random-effects model used (e.g., DerSimonian-Laird, Bayesian).

Results:

• Figure 1: Ensure the PRISMA flowchart follows the original published template, as the current version appears to be a modified and non-standard representation. Provide a more detailed description of the “other sources” section, including the inclusion and exclusion process. Carefully check for typos (e.g., inconsistent use of parentheses and n=).

• Table 1: The numbers presented in the prevalence column are not percentages and do not accurately represent prevalence. Please clarify this, as the current format is incorrect and highly confusing.

• Lines 183-184: Explain how more than half of the population is classified as “hospital-based” if the inclusion criterion is healthy populations.

• Line 185: It is not necessarily required for the study design to be explicitly stated; it can often be identified from the described methodology. Please clarify why these studies were classified as having an unknown study design.

• Lines 187-188: Include references for the studies mentioned.

• Line 194: Revise the vague term “A maximum number.”

• Line 197: Index S2 Table in the text by stating “in Table 1 and S2 Table.”

• Line 257: Correct the indexing error; supplementary figure 5 is for the forest plot of HSV among males, while the results of the quality assessment are in supplementary table 3.

General Comments:

1. Major concerns regarding the inclusion and exclusion criteria need to be addressed, as they appear overly restrictive.

2. Figures are of poor resolution and must be improved for publication-quality presentation.

Reviewer #2: The authors should be commended for their effort in coming up with this paper. I have found the document to be technically sound, although it would benefit from extensive review of typographical and grammatical/language errors some of which I have highlighted in the attached revised article.

**Do you want your identity to be public for this peer review?** For information about this choice, including consent withdrawal, please see our Privacy Policy

Reviewer #1: No

Reviewer #2: **Yes: ** Ayodeji O. OLARINMOYE

---

## [Decision Letter · Decision Letter 1]

26 Oct 2025

PGPH-D-24-02429R1

Prevalence of CMV, EBV, HPV, and HSV among South Asian Healthy Population: A Systematic Review and Meta-Analysis

Dear Dr. Ahmed,

Thank you for submitting your manuscript to PLOS Global Public Health. After careful consideration, we feel that it has merit but does not fully meet PLOS Global Public Health’s publication criteria as it currently stands. Therefore, we invite you to submit a revised version of the manuscript that addresses the points raised during the review process.

The manuscript has been evaluated by one reviewer, and their comments are available below.

In addition, there are a number of concerns remaining over the reporting of the methodology. Your previous response indicated that only keyword searches were utilised, but the manuscript still refers to MeSH terms. In addition, justification of the inclusion and exclusion criteria, as well as the search period, are missing from the manuscript. 

Could you please revise the manuscript to carefully address the concerns raised?

We look forward to receiving your revised manuscript.

Kind regards,

Jen Edwards

Staff Editor

Journal Requirements:

Additional Editor Comments:

There are a number of concerns remaining over the reporting of the methodology. Your previous response indicated that only keyword searches were utilised, but the manuscript still refers to MeSH terms. In addition, justification of the inclusion and exclusion criteria, as well as the search period, are missing from the manuscript. 

Reviewers' comments:

Reviewer's Responses to Questions

**Comments to the Author**

Reviewer #2: (No Response)

publication criteria?

Reviewer #2: Yes

3. Has the statistical analysis been performed appropriately and rigorously?

Reviewer #2: Yes

4. Have the authors made all data underlying the findings in their manuscript fully available (please refer to the Data Availability Statement at the start of the manuscript PDF file)?

Reviewer #2: Yes

5. Is the manuscript presented in an intelligible fashion and written in standard English?

Reviewer #2: No

Reviewer #2: Concerns previously raised regarding how some figures (i.e. Roman Numerals) were written, and the need for extensive language editing, have not yet been addressed by the authors in the revised manuscript R1.

**Do you want your identity to be public for this peer review?** For information about this choice, including consent withdrawal, please see our Privacy Policy

Reviewer #2: **Yes: ** Ayodeji O. OLARINMOYE

---

## [Decision Letter · Decision Letter 2]

11 Dec 2025

Prevalence of CMV, EBV, HPV, and HSV among South Asian Healthy Population: A Systematic Review and Meta-Analysis

PGPH-D-24-02429R2

Dear Mr. Ahmed,

We are pleased to inform you that your manuscript 'Prevalence of CMV, EBV, HPV, and HSV among South Asian Healthy Population: A Systematic Review and Meta-Analysis' has been provisionally accepted for publication in PLOS Global Public Health.

Best regards,

Julia Robinson

Executive Editor

Reviewer Comments (if any, and for reference):

Reviewer's Responses to Questions

**Comments to the Author**

Reviewer #2: All comments have been addressed

publication criteria?

Reviewer #2: Yes

3. Has the statistical analysis been performed appropriately and rigorously?

Reviewer #2: Yes

4. Have the authors made all data underlying the findings in their manuscript fully available (please refer to the Data Availability Statement at the start of the manuscript PDF file)?

Reviewer #2: No

5. Is the manuscript presented in an intelligible fashion and written in standard English?

Reviewer #2: Yes

Reviewer #2: The article is well-written It reports on a group of infectious diseases of global public health concern. Some grammatical errors were observed in the R-2 manuscript and these should be corrected. The authors are advised to proofread the entire manuscript and to engage the services of an English language editor to correct the grammatical issues.

**Do you want your identity to be public for this peer review?** For information about this choice, including consent withdrawal, please see our Privacy Policy

Reviewer #2: No
